# Methodological and Practical Challenges in Synthesizing Occupational Cancer Studies

**DOI:** 10.3390/ijerph21060742

**Published:** 2024-06-06

**Authors:** Soyeon Ahn, Laura A. McClure, Paulo S. Pinheiro, Diana Hernandez, Devina J. Boga, Henna Ukani, Jennifer V. Chavez, Jorge A. Quintela Fernandez, Alberto J. Caban-Martinez, Erin Kobetz, David J. Lee

**Affiliations:** 1Department of Educational and Psychological Studies, School of Education and Human Development, University of Miami, Coral Gables, FL 33146, USA; 2Department of Public Health Sciences, University of Miami Miller School of Medicine, Miami, FL 33101, USA; mcclul2@sage.edu (L.A.M.); ppinheiro@med.miami.edu (P.S.P.); drh118@miami.edu (D.H.); djd132@med.miami.edu (D.J.B.); hxu35@miami.edu (H.U.); j.chavez8@miami.edu (J.V.C.); acaban@med.miami.edu (A.J.C.-M.); ekobetz@med.miami.edu (E.K.); dlee@med.miami.edu (D.J.L.); 3Sylvester Comprehensive Cancer Center, University of Miami Miller School of Medicine, Miami, FL 33101, USA; 4University of Miami Libraries, University of Miami, Coral Gables, FL 33146, USA; jaq32@miami.edu; 5Department of Medicine, University of Miami Miller School of Medicine, Miami, FL 33101, USA

**Keywords:** meta-analysis, cancer studies, mortality, incidence, ICD code

## Abstract

Studies examining occupational exposures and cancer risk frequently report mixed findings; it is thus imperative for researchers to synthesize study results and identify any potential sources that explain such variabilities in study findings. However, when synthesizing study results using meta-analytic techniques, researchers often encounter a number of practical and methodological challenges. These challenges include (1) an incomparability of effect size measures due to large variations in research methodology; (2) a violation of the independence assumption for meta-analysis; (3) a violation of the normality assumption of effect size measures; and (4) a variation in cancer definitions across studies and changes in coding standards over time. In this paper, we first demonstrate these challenges by providing examples from a real dataset collected for a large meta-analysis project that synthesizes cancer mortality and incidence rates among firefighters. We summarize how each of these challenges has been handled in our meta-analysis. We conclude this paper by providing practical guidelines for handling challenges when synthesizing study findings from occupational cancer literature.

## 1. Introduction

Approximately 1.8 million people were diagnosed with cancer in 2019 in the United States (US) [1], and cancer is the second leading cause of death nationwide [2]. Cancer incidence and mortality vary widely by demographic factors, socio-economic status, and region [3]. Worldwide, occupational carcinogenic exposures are responsible for an estimated 13.5 deaths per 100,000 workers [4].

Numerous observational epidemiologic studies demonstrate that the link between cancer incidence and mortality risk varies among occupations [5,6,7], potentially influenced by distinct types of cancer-causing agents. Sedentary office jobs can heighten the risk of cancer due to behavioral factors, such as physical inactivity [8]. Other workers are directly/indirectly exposed to known or suspected cancer-causing agents, such as sunlight [9], hazardous chemicals [10], and infectious agents [11,12], in their work environment [13]. For example, the IARC has classified painting, firefighting [14], and shiftwork as occupations with increased cancer risk [14,15]. However, these cancer-causing exposures vary widely depending on job tenure, specific job duties, and enforcement of safety practices.

Therefore, this growing body of occupational cancer literature often shows mixed results that can be attributed to various factors such as missing information, variation in defining cancer risk exposure, and the use of different comparison groups; it is thus imperative for researchers to synthesize the literature regarding cancer incidence and mortality and further identify any potential explanations for the observed variabilities in findings. However, when conducting meta-analyses, practical and methodological challenges are often encountered. In this study, we demonstrate some of those challenges using a real meta-analytic dataset collected to estimate cancer risk in firefighters. Then, we discuss how those challenges have been handled in our own meta-analysis [16], where detailed information related to our data collection and analysis can be found. We conclude by providing some practical guidelines for handling those challenges when synthesizing occupational cancer studies. 

## 2. Methodological and Practical Challenges in Synthesizing Study Findings from Occupational Cancer Studies 

There are a number of challenges that may be encountered when synthesizing findings from occupational cancer studies; however, we endeavor to address the following: (1) incomparability of effect size measures due to large variation in research methodology; (2) violation of the independence assumption for synthesizing effect size measures extracted from individual studies; (3) violation of the normality assumption for effect size measures; and (4) inconsistent use of the standards for diagnosing and coding cancer over time across studies.

### 2.1. Comparability of Effect Size Measures Due to Variation in Research Methodology

In occupational cancer studies, an investigator observes which outcome occurs upon exposure to risk factors, describes the probability of that outcome, and analyzes the magnitude of the association between risk factors and the outcome. To perform this, researchers employ different study designs, most frequently retrospective, case–control, and cross-sectional designs. Of sixty-five primary studies examining cancer incidence and mortality among firefighters for our meta-analysis [16], there are three cross-sectional, nineteen case–control, and forty-three cohort studies. Not only do these studies differ in the research questions being asked, but also, they use a different set(s) of effect size measures that quantify the relationship between risk factors and the outcome. 

In retrospective studies, the comparison of outcome between groups exposed to specific risk factors and a non-exposed comparison group often relies on a simplistic 2 × 2 table (exposed vs. not exposed) (observed outcome or not), without considering the continuum of exposure and their relationship to the outcome. Here, researchers use various effect size measures such as relative risk (RR, also called risk ratio, rate ratio, and relative rate ratio), the hazard ratio (HR), the standardized incidence ratio (SIR), the standardized mortality ratio (SMR), the proportional mortality ratio (PMR), and others. For example, Daniels et al. [17] examined cancer incidence and mortality in a cohort of 29,993 US firefighters employed since 1950 and reported both SIRs and SMRs comparing firefighters to the general US population. Another cohort study conducted by Baris et al. [18] provided SMRs and RRs for Philadelphia firefighter cancer deaths between 1925 and 1986. In contrast, Coggon et al. [19] used PMRs for British firefighter cancer deaths. It is, therefore, evident that even among studies using the same study design, the multitude of statistical information provided makes direct comparisons difficult across studies.

In case–control studies, wherein two groups of people—a group of people with the condition or disease (cases) and a comparable group of people without the condition or disease (controls)—are compared, an odds ratio (OR) is frequently used. The OR is a relative measure quantifying the ratio of two groups of odds that an outcome occurs given the presence/absence of a particular exposure. Because an OR quantifies two relative odds between groups, its comparability depends on how genuinely comparable the two groups are across studies. For example, the ratios of the firefighters’ odds of being diagnosed with a certain type of cancer are different, depending on the other occupational groups to which they are being compared, such as the general population [20], policemen [21,22,23], indoor office workers [24], or military personnel [25]. Further, some studies use other cancer cases as the control group [26]. Therefore, special attention is required when interpreting study findings and combining effect size measures in a meta-analysis. Practically, instead of combining all ORs in a meta-analysis, regardless of the comparison group, we strongly recommend performing either a subgroup analysis (also known as a moderator analysis) or aggregating effect size measures separately according to the comparison group. 

In cross-sectional studies focusing on outcomes for one occupational group (in this case, firefighters) at one point in time or over time, ORs or RRs are most commonly used to compare frequencies of outcomes by different moderating factors such as gender, years of experience, smoking status, and race/ethnicity. Moore and colleagues [27] reported ORs for firefighter skin cancer incidence associated with age, gender, ethnicity, tanning bed use, and years of working experience. Moreover, no two studies in our dataset adjusted for identical covariates. For instance, Karami and her colleagues [28] used sex, age at reference date, race, study center, education level, history of hypertension, smoking status, Body Mass Index, and family history of cancer as covariates. On the other hand, Bates and Lane [29] used age, year of diagnosis, ethnicity, and an indicator of socio-economic status to predict cancer incidence. 

Likewise, when synthesizing observational epidemiological studies in practice, meta-analysts often observe considerably large variations in the (1) study design employed, (2) type of effect size measures, (3) statistical information reported, and (4) research questions. These variations raise concerns regarding the comparability of study findings and the most appropriate effect size measure to choose. More importantly, unless theoretical or methodological justification exists for converting among the common effect size metrics [30], different types of effect size measures (e.g., SMR and SIR) must be analyzed in a separate set of meta-analyses. Furthermore, separate meta-analyses should be performed, depending on which comparison group was used and/or what sociodemographic or health variables were included as covariates. 

### 2.2. Dependency Issues in Effect Sizes Arising from Data Overlapping in Time and Regions

The primary studies in a meta-analysis often include overlapping data when populations come from the same geographic region and/or during the same time period. This creates dependency issues in statistical analysis, violating the independence assumption for a univariate meta-analytic technique. For instance, two case–control studies examining cancer mortality in California—one with data collected between 1988 and 2003 [31] and the other with data collected between 1988 and 2007 [32]—showed 15 years of overlap in data collection. Thus, during the overlap period, it is likely that participants were counted more than once when data were pooled while ignoring such dependency. A similar example can be seen in two studies conducted in Nordic countries—one between 1961 and 2005 [33] and the other between 1968 and 2014 [25]. The overlap period here includes most of the years of the two studies. 

Figure 1 and Figure 2 display the number of studies published between 1980 and 2019 that examined firefighter cancer incidence and mortality rates from the same geographic regions—in US states (Figure 1) and in European countries (Figure 2). As shown in Figure 1, a large number of effect sizes (*k* = 120) were from California (*k* = 10), New York (*k* = 6), New Jersey (*k* = 5), Illinois (*k* = 5), Pennsylvania (*k* = 4), Washington (*k* = 4), and Florida (*k* = 4). This is likely because either the cancer registry data have been available for a much longer period in these states compared to other states or because these states have a large population and thus an enhanced ability to detect cancer risk associations. Similarly, as shown in Figure 2, particular countries in Europe, such as France (*k* = 12), Denmark (*k* = 8), Sweden (*k* = 6), Iceland (*k* = 5), and Finland (*k* = 6), produced multiple studies reporting firefighter cancer incidence and mortality rates due, in part, to robust surveillance systems available in these countries. 

When multiple studies with overlapping data are included in a meta-analysis (as shown in Figure 3), dependency reduces the accuracy of overall effect size estimates and their statistical inference. In particular, the point estimate may be biased in the direction of an effect size that is included several times. Our simulation results with 1000 replications (see Figure 4), where different percentages of dependent effects were meta-analyzed, suggest very little bias in the meta-analytic estimates (i.e., less than |0.01|) and upwardly biased standard errors (see Figure 5). Bias in the meta-analytic estimates and their standard errors were similar, regardless of the percentage of overlapping studies included in a meta-analysis (i.e., 15%, 30%, 45%, and 60%). 

Likewise, these dependent effect sizes artificially inflate the associated standard error as they have less variation. The magnitude of effect sizes extracted from overlapping time periods or regions could also increase the likelihood of committing type I errors, thus lowering the statistical power of inferences. Specifically, including particularly large effect sizes increases the risk of committing more type I errors by biasing the point estimation further away from the true population value and further deflating the associated standard error. Conversely, if primary studies demonstrating small effect sizes overlap with one another, underpowered meta-analytic study findings will be driven by underestimated point estimation values. 

In practice, these issues often occur, yet are ignored in many published meta-analytic reviews. One systematic review of 60 meta-analyses in health sciences found that more than half reported dependencies stemming from overlapping information in effect size computation, with a median of 25% of the primary studies having been included more than once across meta-analyses. The remaining meta-analyses did not even address such dependence issues [34]. This suggests that meta-analytic research studies in practice may be fundamentally imprecise and methodologically weak, and are, therefore, not accurate representations of true phenomena. 

Although no widely accepted method exists for handling dependency resulting from overlapping information, methodologists suggest two possible solutions: (1) excluding findings from redundant studies, and (2) for studies where redundancy is unavoidable (e.g., exclusion of findings is not plausible), applying advanced statistical methods that can adjust for the impact of redundancy (e.g., a multilevel meta-analysis [35] or a robust variance estimation method [36]). These methods are an improvement over current practice but are not foolproof, as they require researchers to impute an unknown value to quantify the degree of dependence in effect size measures (i.e., intraclass correlation coefficient). As no single current method is completely satisfactory, methodological advances must be developed to offer practical guidance to researchers for handling the issue of dependency due to overlapping information. 

### 2.3. Violation of Normality Assumption for Synthesizing Effect Size Measures

For rare outcomes or diseases, the distribution of the observed effect size will not follow a normal distribution, with a large value of standard error and, consequently, a wider 95% confidence interval (CI). For instance, Demers et al. [20] examined cancer incidence in a cohort of 2447 male firefighters in Seattle and Tacoma from 1974 to 1989. In comparison to the general male population, the reported SIR of firefighter sinus cancer was 2.2 (i.e., observed *n* = 1 vs. expected *n* = 0.5), with a 95% CI of 0.1–12.4, which was not symmetric, mainly due to the large standard error associated with the SIR. The same pattern is found in other cancers (e.g., SIR = 2.4, 95% CI = 0.1–13.3 [observed *n* = 1 vs. expected *n* = 0.4] for male firefighter breast cancer; SIR = 0.8, 95% CI = 0.2–4.2 [observed *n* = 1 vs. expected *n* = 1.3] for male firefighter thyroid cancer). These asymmetric patterns in 95% CIs around the observed SIR differ from that of more common cancers, which are approximately normal and symmetric around the mean effect size value (i.e., SIR = 1.1, 95% CI = 0.7–1.6 [observed *n* = 23 vs. expected *n* = 21] for firefighter colon cancer; SIR = 1.0, 95% CI = 0.7–1.3 [observed *n* = 45 vs. expected *n* = 46.8] for firefighter lung, trachea, bronchus cancer). 

Although a number of simulation studies have been performed to use other effect size measures, such as Hedge’s *g* or Cohen’s *d*, researchers have shown that violations of non-normality of effect sizes included in a meta-analysis have various adverse impacts on overall meta-analytic results, demonstrating poor performance for a point estimation [37] and its associated confidence intervals and inflating type I error rates of the *Q* test [38,39]. This pattern occurs even when the average sample size and the number of effect sizes included in a meta-analysis are large [40]. For such cases, therefore, it is strongly recommended that the traditional meta-analytic method, based on the normality assumption for effect size measures, should not be used. 

Considering that, we have conducted our own comparison study between the outcomes of traditional meta-analysis and Bayesian meta-analysis, taking into account the violation of assumptions regarding the normal distribution in the latter. Our findings indicate that traditional meta-analysis yielded a higher number of statistically significant average SMR or SIR values compared to Bayesian meta-analysis. Moreover, while the point estimate values of the SMR or SIR exhibited considerable similarity, the confidence intervals surrounding these estimates were notably wider in Bayesian meta-analysis as opposed to traditional meta-analysis. Given that little is known about the robustness of meta-analytic results of synthesizing non-normal SMRs or SIRs, the most used effect size measures in occupational cancer studies, further research is imperative. 

### 2.4. Inconsistencies in Cancer Diagnoses over Time

Another methodological challenge in estimating cancer risk across studies is the change in cancer coding over time. All but the very oldest studies use a coding system called the International Classification of Diseases (ICD) (https://www.who.int/classifications/icd/en/ (accessed on 1 June 2024), which is periodically revised to include the addition of new diseases or subgroupings or to add detail to current disease classifications. While helpful in further refining cancer diagnosis and classification, these changes make it difficult to uniformly code cancers for the purposes of meta-analysis.

There are eleven versions of the ICD, used mostly for mortality, and three versions of the ICD specific to cancer (ICD for Oncology (ICD-O)), used for incidence. A crosswalk, which converts from one version to the previous, is created for each new version of the ICD, but there is no master crosswalk for all versions, which would, of course, vary, depending on the version used as the reference. For many cancer sites, there is a one-to-one coding (e.g., prostate), but for others, there are one-to-many or many-to-one differences across ICD versions (e.g., lymphoma). There are situations where new cancer sites are added that were not present in previous versions, such as Kaposi’s sarcoma, which does not exist in ICD-6 or earlier. Additionally, certain cancers, including mesothelioma, head and neck cancer, and specific types of lymphomas and leukemias, vary widely in occurrence over time and across different global regions. As a result, the estimates of incidence rates by the World Health Organization might be lower than actual figures due to insufficient cancer registries and difficulties in accessing diagnostic facilities. Therefore, any studies using these older versions could not be included in a meta-analysis examining this cancer. Additionally, cancers that may have in fact been Kaposi’s sarcoma at the time of earlier coding schemes would have been coded as something less specific, with perhaps little consistency in coding. There are similar new cancer diagnoses that were included in ICD-10 that were not included in any previous version.

Much of the current research in cancer that is key for contemporary meta-analyses has coded cancer sites using ICD-9, ICD-10, or ICD-O-3, for which crosswalks exist. Even so, information can be lost when the earlier version includes less detail than the later version, which is a concern when using earlier versions as the reference. Alternatively, there may be challenges in using later versions as the reference, as is the case for lymphomas, where cancer was considered to be of a certain classification in the past but was later discovered to not be a distinct diagnosis and thus coded more generally. In this case, meta-analysts run the risk of losing some nuance related to the original coding.

Despite these challenges in reconciling coding schemes, some publications do not specify which specific codes or even which ICD version was used. Meta-analysts are, therefore, reliant on the names of the cancer sites used in the study, which is problematic as naming systems change over time just as codes do. So here, also, information may be lost when making assumptions about the author’s definition of a particular cancer site. 

To better address these complexities, we recommend that an expert panel should be convened to create and publish a crosswalk covering all ICD coding versions that can then be used by all investigators conducting cancer meta-analytic studies. In the meantime, we recommend that researchers provide a full description of their crosswalk process of classifying cancer codes for their meta-analyses, which is imperative for conducting a reproducible, replicable, and generalizable systematic and meta-analytic review. 

## 3. Discussion

Before combining all effect size measures across a set of included studies, researchers’ careful investigations of these salient issues are strongly recommended. In addition, acknowledging that the current meta-analytic practices do not provide one single and widely accepted solution to address all challenges, researchers are encouraged to fully explain their rationale for any subjective judgments being made throughout their meta-analytic studies. 

First, it is crucial to recognize the necessity of theoretical or methodological justification when converting among common effect size metrics. Without such justification, different types of effect size measures, such as standardized mortality ratios (SMRs), standardized incidence ratios (SIRs) and odds ratios (ORs), should be analyzed separately in meta-analytic studies. This ensures the integrity and accuracy of the analysis by preventing inappropriate synthesis of significantly disparate pieces of information. Moreover, the complexity of the data warrants separate meta-analyses based on various factors, including the comparison group utilized and the inclusion of sociodemographic or health variables as covariates. By conducting separate meta-analyses for distinct subsets of data, researchers can more effectively capture the nuances of the underlying relationships and provide insights that are both comprehensive and nuanced. This approach enhances the robustness and validity of meta-analytic findings, thereby contributing to a better understanding of the phenomena under investigation.

Second, despite well-established guidelines cautioning against synthesizing dependent effect sizes, a significant number of meta-analyses in health sciences overlook this critical aspect. Alarmingly, a considerable portion of these meta-analyses failed to address such dependence issues altogether. Additionally, the violation of assumptions regarding normality is a significant concern, particularly in the context of rare diseases or when samples lack representation from specific populations, such as women firefighters. This trend underscores a concerning reality: many meta-analytic research studies may suffer from fundamental imprecision and methodological weaknesses, thereby compromising their accuracy and reliability as representations of true phenomena. It highlights the urgent need for adherence to recommended methodologies to ensure the validity and credibility of research findings.

Lastly, the challenges inherent in reconciling coding schemes and the lack of specificity in publications concerning cancer coding and naming systems present multifaceted implications. First, there is a significant risk of misinterpretation. The absence of clear specifications regarding coding schemes and versions heightens the potential for meta-analysts to misinterpret data. Evolving naming systems and coding versions can result in differing interpretations of cancer sites, leading to inconsistent findings and erroneous conclusions. Secondly, there is a loss of critical information. The failure to provide explicit coding information can result in the loss of vital data for meta-analyses. Meta-analysts may struggle to fully understand how cancer sites were classified, resulting in incomplete or inaccurate synthesis of evidence. Moreover, concerns regarding reproducibility and replicability arise. The lack of transparency in the classification process undermines the reproducibility and replicability of meta-analytic studies. Without detailed documentation of the crosswalk process, replicating the findings or verifying the analysis’s robustness becomes challenging for other researchers. Therefore, the necessity for standardization in cancer coding practices is evident. The recommendation to convene an expert panel to establish a comprehensive crosswalk for all ICD coding versions emphasizes this need. A standardized crosswalk would promote consistency across studies and enhance the comparability of results. Furthermore, methodological rigor is essential. Providing a thorough description of the crosswalk process is pivotal for ensuring methodological rigor in meta-analytic reviews. Transparency in the classification methodology enables reviewers to assess the validity and reliability of the findings, thereby contributing to the overall quality of the research.

Furthermore, with the increased use of meta-analytic reviews in the medical field, we raise concerns about the quality of meta-analyses and thus suggest assessing the quality of published meta-analyses. A number of tools [41,42,43,44] can be used to assess the quality of the published meta-analyses (i.e., cross-over trials, double counting, statistical heterogeneity, comparability of comparison). In our meta-analysis [16], the meta-analytic result was found to be significantly increased by 29%, for every additional increase in the quality score of the included studies. Because these methodological and practical challenges may negatively affect the reproducibility, replicability, and generalizability of findings, researchers must carefully evaluate the appropriateness of data collection, data analysis, and data interpretation when conducting a systematic and meta-analytic review. 

## 4. Conclusions

This study highlights significant methodological and practical challenges encountered when synthesizing data from a collection of epidemiological studies using a real dataset collected for a meta-analysis of firefighter cancer incidence and mortality rates. 

Challenges discussed in this paper often arise from (1) incomparability of effect size measures due to study design, statistical information reported, research question, and choice of effect size measure, (2) variability observed in a set of included studies, (3) violation of the underlying assumption for meta-analytic techniques, normality and independence of effect size measures, and (4) variation in cancer definitions across studies and changes in coding standards over time.

To address these challenges, our recommendations include (1) refraining from analyzing different types of effect size measures together; (2) meticulously examining potential dependencies and adjusting for them in subsequent statistical analyses; (3) employing statistical methods that do not rely on the normality assumption; and (4) integrating an expert panel for careful cross-referencing of diverse cancer definitions. 

## Figures and Tables

**Figure 1 ijerph-21-00742-f001:**
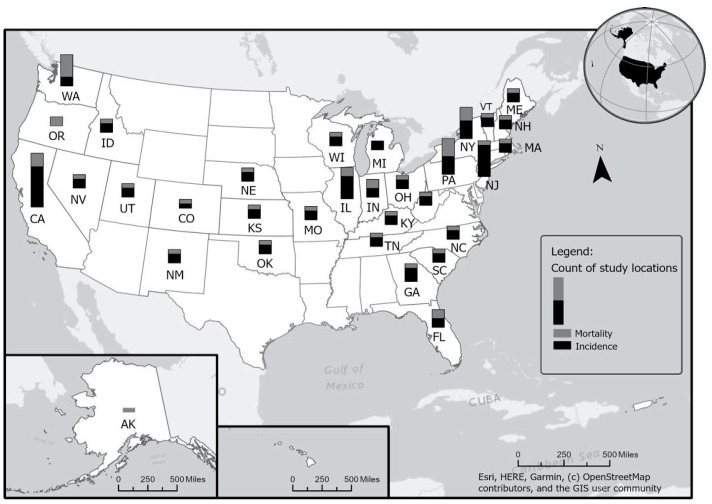
Number of overlapping studies in the US. See Appendix A for the studies included.

**Figure 2 ijerph-21-00742-f002:**
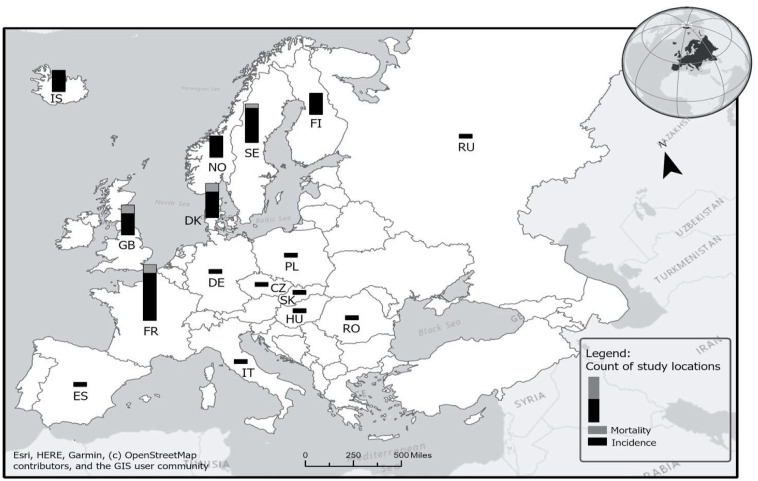
Number of overlapping studies in Europe. See Appendix A for the studies included.

**Figure 3 ijerph-21-00742-f003:**
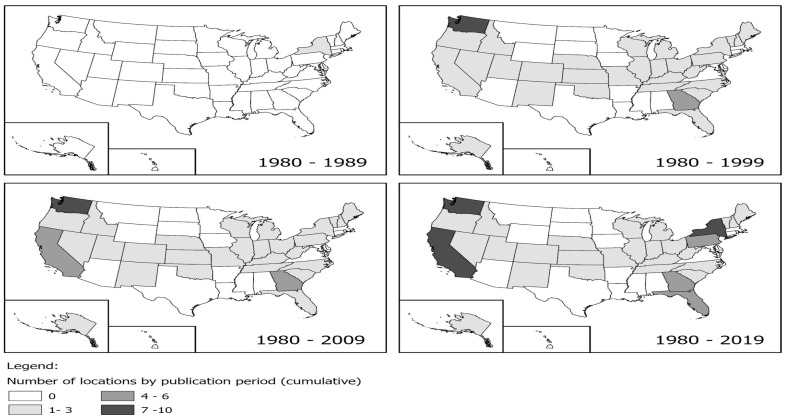
Number of overlapping studies over time in the US.

**Figure 4 ijerph-21-00742-f004:**
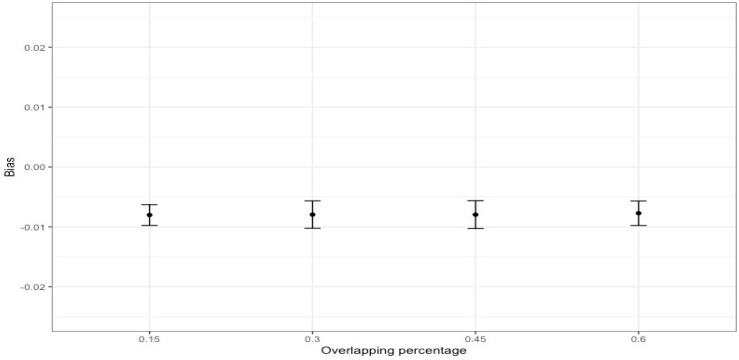
Bias of meta-analytic effects by the percentage of overlapping studies.

**Figure 5 ijerph-21-00742-f005:**
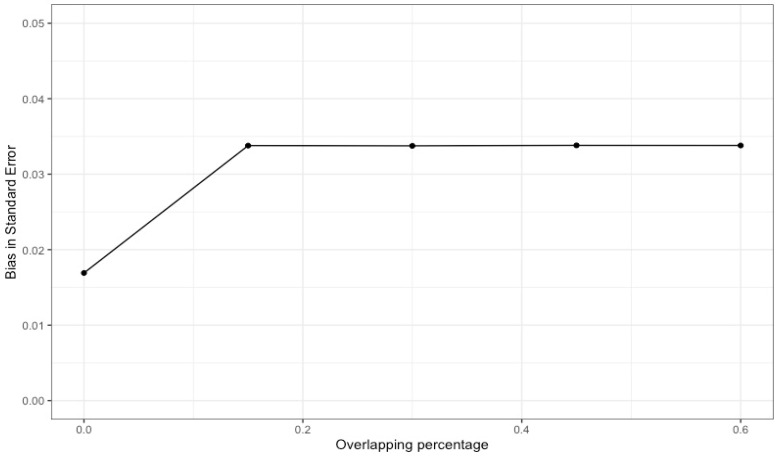
Bias in the standard error of meta-analytic effects by the percentage of overlapping studies.

## Data Availability

The data presented in this study are available in the Appendix A.

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
