# Peer review of "Methodological and Practical Challenges in Synthesizing Occupational Cancer Studies"

_ijerph, 2024, doi:10.3390/ijerph21060742_

Round 1

Reviewer 1 Report (Previous Reviewer 2)

Comments and Suggestions for Authors

The new version fully answers to all observations about the primary versione.

Please, check the lines 139-140, I'm not sure that the sentence Is correctly written (sorry, I forgot this point at the primary review).

Line 338: "furthermore".

Author Response

We appreciate the editor and reviewers' feedback. We revised the text to address their concerns and marked it in red. We are confident that our changes are adequate, but we can explain more if necessary.

Thank you.

Reviewer1

The new version fully answers to all observations about the primary version.

Thank you!

Please, check the lines 139-140, I'm not sure that the sentence Is correctly written (sorry, I forgot this point at the primary review).

We appreciate you noticing this error. We fixed it in Line 131-132

Line 338: "furthermore".

We appreciate you noticing this error. We fixed it in Line 330.

Reviewer 2 Report (New Reviewer)

Comments and Suggestions for Authors

The authors have outlined four types of problems in meta-analyses that may lead to error.  They give instances of each of these problems encountered in their recently-published meta-analysis of cancer studies in firefighters (their reference 17). These challenges are clearly explained, and illustrated by reference to a meta-analysis of cancer in firefighters by the same authors. However it is relevant to note that an IARC Working Group has also recently published a meta-analysis of multiple cancer types in firefighters, in which the results convinced the Working Group that firefighters have excess bladder cancers, whereas the study by these authors (reference 17) indicates otherwise.  This disparity suggests that such meta-analyses fail to provide definitive answers, even when the pitfalls outlined in this study are addressed. 

Both of these studies address all cancer types, so that the challenges of meta-analyses are further complicated by the effects of multiple comparisons, in which a statistically significant result will occur so often purely by chance and highlighted (while estimates significantly below the null are often not). It may be preferable to restrict meta-analyses to measurement of the associations between a particular exposure and a single disease entity. 

Most studies of occupational cancer address a particular exposure, with an a priori hypothesis based on reasons for a possible causal link to a particular cancer. Studies of cancer in firefighters are different in that no specific exposure is addressed, so that a crucial source of error is not discussed - the need to ensure that all studies included in a meta-analysis are of subjects who were exposed, and preferably heavily exposed.  If a causal link between an exposure and a cancer exists, it is most likely to be identified from studies of highly exposed subjects.  Conversely care must be taken to exclude studies where exposures were either low or absent.   An example of this principle can be found in a study of benzene and non-Hodgkin lymphoma (NHL) by Vlaanderen et al, where the Relative Risk of lymphoma was estimated in a subgroup of studies only where there was an excess of acute myeloid leukaemia (AML).  Since benzene is is a known cause of AML, study populations in which no excess was detected were less likely to have had sufficient exposure to benzene to cause a detectable increase in NHL, and were therefore excluded.  This important source of underestimating risk is not addressed in studies of firefighters since no specific exposure is identified or measured.

In summary, the authors have identified a number of pitfalls in meta-analyses and explained how they may be addressed, but they may have been better illustrated by reference to a study or studies of individual cancers and measurable exposures.

34-43.  This section is highlighted in yellow (as are other parts of the text), suggesting that it has been inserted on the suggestion of a prior reviewer. It is not germane to the topic and should be omitted.

44-45.  The point of this sentence is unclear.  That there is a link between cancer incidence and mortality is obvious, since many cancers are fatal. The link between them varies with the cancer (eg most skin cancers are not fatal) but it is less clear that it varies with occupation and the three cited references do not address that question. 

134.  I would suggest an alternative heading.  The section addresses the problem of overlapping, ie where some subjects are common to two or more studies included in a meta-analysis.  The heading Dependency in Effect-Size Measures doesn’t really convey that meaning. Nevertheless the discussion on quantifying the effect of overlapping is informative and interesting.

Figures 1-3 do not add anything and could be deleted (although this doesn’t matter much in an electronic publication).

297.  There appears to be a word missing after “appropriate”.

338 “Furthermore” is mis-spelt.

Author Response

We appreciate the editor and reviewers' feedback. We revised the text to address their concerns and marked it in red. We are confident that our changes are adequate, but we can explain more if necessary.

Thank you.

The authors have outlined four types of problems in meta-analyses that may lead to error.  They give instances of each of these problems encountered in their recently-published meta-analysis of cancer studies in firefighters (their reference 17). These challenges are clearly explained, and illustrated by reference to a meta-analysis of cancer in firefighters by the same authors. However it is relevant to note that an IARC Working Group has also recently published a meta-analysis of multiple cancer types in firefighters, in which the results convinced the Working Group that firefighters have excess bladder cancers, whereas the study by these authors (reference 17) indicates otherwise.  This disparity suggests that such meta-analyses fail to provide definitive answers, even when the pitfalls outlined in this study are addressed. 

We appreciate the comments. We added an IARC Working Group to our paper, showing that Firefighters have a higher occupational risk.

We also want to note that different meta-analyses can have different results if they use different studies, unless there is a lot of overlap. The working group did not rely only on meta-analyses for causality. They chose a specific list of studies for their meta-analysis, as below:

“Several cohort studies of firefighters compared with the general population consistently showed positive associations for bladder cancer incidence. The Working Group’s meta-analysis of ten studies had a small (16%) but precise and consistent increased risk estimate (95% CI 8–26%, I²=0).”

Both of these studies address all cancer types, so that the challenges of meta-analyses are further complicated by the effects of multiple comparisons, in which a statistically significant result will occur so often purely by chance and highlighted (while estimates significantly below the null are often not). It may be preferable to restrict meta-analyses to measurement of the associations between a particular exposure and a single disease entity. 

We appreciate this comment. However, we want to point out that the working group was very careful in choosing which cancers were included in the meta-analysis. It is correct that combining multiple cancers in occupational meta-analyses can raise the possibility of random findings. Moreover, restricting the analysis to a specific exposure and a single disease type might work well for painters who have similar exposures but not for firefighters, and maybe other occupational groups.

Most studies of occupational cancer address a particular exposure, with an a priori hypothesis based on reasons for a possible causal link to a particular cancer. Studies of cancer in firefighters are different in that no specific exposure is addressed, so that a crucial source of error is not discussed - the need to ensure that all studies included in a meta-analysis are of subjects who were exposed, and preferably heavily exposed.  If a causal link between an exposure and a cancer exists, it is most likely to be identified from studies of highly exposed subjects.  Conversely care must be taken to exclude studies where exposures were either low or absent.   An example of this principle can be found in a study of benzene and non-Hodgkin lymphoma (NHL) by Vlaanderen et al, where the Relative Risk of lymphoma was estimated in a subgroup of studies only where there was an excess of acute myeloid leukaemia (AML).  Since benzene is is a known cause of AML, study populations in which no excess was detected were less likely to have had sufficient exposure to benzene to cause a detectable increase in NHL, and were therefore excluded.  This important source of underestimating risk is not addressed in studies of firefighters since no specific exposure is identified or measured.

We agree that this is an important topic although less germane in the study of firefighters with their complex carcinogenic exposure profiles.  In studies of other occupational groups looking at dose-response associations would be preferable, but many studies lack such information and investigators are forced to relay of proxies such as years of service. 

In summary, the authors have identified a number of pitfalls in meta-analyses and explained how they may be addressed, but they may have been better illustrated by reference to a study or studies of individual cancers and measurable exposures.

We agree with your observation. That is why our study will provide useful insights on this variation and inform future meta-analysis, while paying close attention to various issues we raised in this study.

34-43.  This section is highlighted in yellow (as are other parts of the text), suggesting that it has been inserted on the suggestion of a prior reviewer. It is not germane to the topic and should be omitted.

Thank you for suggestion – we deleted it.

44-45.  The point of this sentence is unclear.  That there is a link between cancer incidence and mortality is obvious, since many cancers are fatal. The link between them varies with the cancer (eg most skin cancers are not fatal) but it is less clear that it varies with occupation and the three cited references do not address that question. 

Thank you for your feedback. We have revised to this sentence in line 35 – 37. 

  1. I would suggest an alternative heading.  The section addresses the problem of overlapping, ie where some subjects are common to two or more studies included in a meta-analysis.  The heading Dependency in Effect-Size Measures doesn’t really convey that meaning. Nevertheless the discussion on quantifying the effect of overlapping is informative and interesting.

Thank you for your feedback. The section title is now changed to “Dependency Issues in Effect Sizes Arising from Data Overlapping in Time and Regions”, shown in Line 126.

Figures 1-3 do not add anything and could be deleted (although this doesn’t matter much in an electronic publication).

Thank you for your feedback. But we believe these figures can show the potential chronological and/or geographical overlaps that cause the dependency issues in effect sizes.

  1. There appears to be a word missing after “appropriate”. We appreciate you noticing this error. We fixed it on Line 298-289.

338 “Furthermore” is mis-spelt.

We appreciate you noticing this error. We fixed it in Line 330.

This manuscript is a resubmission of an earlier submission. The following is a list of the peer review reports and author responses from that submission.

Round 1

Reviewer 1 Report

Comments and Suggestions for Authors

This paper presents the potential issues and limitations of meta-analysis-type studies in the evaluation of occupational related cancers. The authors use the example of cancers observed in firefighters and select multiple studies that examined the type and incidence rates in different countries. The text is well-written and makes for a great read, highlighting the main issues that can occur with meta-analysis and how data can potentially be misleading due to an incorrect methodology. The authors also present possible methods for avoiding such problems. 

However, there are several aspects that can be improved:

- in the methods sections a description of how the studies were selected, time interval and analisys methods employed would be a great addition;

- the conclusions should follow the discussion section

- conclusions should comprise the main ideas that result from this study, I would recommend this paragraph to be rephrased.

Author Response

This paper presents the potential issues and limitations of meta-analysis-type studies in the evaluation of occupational related cancers. The authors use the example of cancers observed in firefighters and select multiple studies that examined the type and incidence rates in different countries. The text is well-written and makes for a great read, highlighting the main issues that can occur with meta-analysis and how data can potentially be misleading due to an incorrect methodology. The authors also present possible methods for avoiding such problems. 

Thank you for your feedback.

However, there are several aspects that can be improved:

- in the methods sections a description of how the studies were selected, time interval and analysis methods employed would be a great addition;

We have referenced our previously published paper, which outlines the selection process, time interval, and analytical methods used in the studies. Refer to line 61-62.

- the conclusions should follow the discussion section

We have placed the conclusion after the discussion section. Refer to line # 285-316.

- conclusions should comprise the main ideas that result from this study, I would recommend this paragraph to be rephrased.

We have revised the conclusion section to align with the results from this study. Please refer to the line # 302-316.

Reviewer 2 Report

Comments and Suggestions for Authors

A robust, very well designed and very well conducted study, providing relevant methodological tools of pratical utility for the amelioration of the meta-analytic research.

Some notes just about the introduction and the chapters 2.1 and 2.5.

Lines 34-37:

- useful to mention a set of other very impacting agents that IARC classified as established carcinogens, such as asbestos, free crystalline silica, formaldehyde, wood dusts, and leather dusts;

- useful to expressly define the meaning of the terms "carcinogenic", "probably carcinogenic" and "possibily carcinogenic" adopted by IARC.   

Lines 47-50: useful to hint to the reasons justifying at least a part of the "mixed results" shown by the occupational cancer literature, starting from the very common problems of: i) a poor definition of the exposure (poor identification of the individual involved agents, the duration of the exposure, the mean intensity of the exposure and its fluctuation over the time); ii) an overlapping between groups of really unexposed subjects and groups of subjects with medium- or low-level doses of cumulative exposure.   

Lines 66-68: useful to note that frequently the epidemiological studies  explore the relationship between the exposure to carcinogenic agents (one or more than one of them, simultaneously  or at different times), and the occurrence of cancer cases and / or of consequential deaths adopting the binomial approach "exposed vs unexposed"; expecially in the presence of large variability of the cumulative doses inside the exposed population and / or in the presence of a relevant percentage of the general population exposed to low levels of "ubiquitous" carcinogens (e.g. benzene, formaldehyde, PAHs) for a relevant part of ther life too, the effect - size measures could be widely altered.

Lines 242-250: useful to mention other cancers whose classification suffers of a large variability in space and time, such as mesotheliomas, some head and neck cancers, some leukemias and lymphomas; useful to mention that in medium- and low-income countries, the incidence of selected neoplasms (e.g. mesotheliomas and some types of leukemias and lymphomas) could be unterestimated following criticities in the diagnostic process.                      

Author Response

A robust, very well designed and very well conducted study, providing relevant methodological tools of pratical utility for the amelioration of the meta-analytic research.

Some notes just about the introduction and the chapters 2.1 and 2.5.

Lines 34-37:

- useful to mention a set of other very impacting agents that IARC classified as established carcinogens, such as asbestos, free crystalline silica, formaldehyde, wood dusts, and leather dusts;

We have included this suggestion and revised it accordingly.  Please refer to line 34 – 43.

- useful to expressly define the meaning of the terms "carcinogenic", "probably carcinogenic" and "possibily carcinogenic" adopted by IARC.   

We have included this suggestion and revised it accordingly.  Please refer to line 34 – 43.

Lines 47-50: useful to hint to the reasons justifying at least a part of the "mixed results" shown by the occupational cancer literature, starting from the very common problems of: i) a poor definition of the exposure (poor identification of the individual involved agents, the duration of the exposure, the mean intensity of the exposure and its fluctuation over the time); ii) an overlapping between groups of really unexposed subjects and groups of subjects with medium- or low-level doses of cumulative exposure.   

We have included explanations for the mixed findings that are often observed in occupational cancer literature.  Please refer to line 53 – 55.

Lines 66-68: useful to note that frequently the epidemiological studies  explore the relationship between the exposure to carcinogenic agents (one or more than one of them, simultaneously  or at different times), and the occurrence of cancer cases and / or of consequential deaths adopting the binomial approach "exposed vs unexposed"; expecially in the presence of large variability of the cumulative doses inside the exposed population and / or in the presence of a relevant percentage of the general population exposed to low levels of "ubiquitous" carcinogens (e.g. benzene, formaldehyde, PAHs) for a relevant part of ther life too, the effect - size measures could be widely altered.

We have incorporated this approach into the analysis commonly observed in epidemiological studies. Please refer to line 84 – 87.

Lines 242-250: useful to mention other cancers whose classification suffers of a large variability in space and time, such as mesotheliomas, some head and neck cancers, some leukemias and lymphomas; useful to mention that in medium- and low-income countries, the incidence of selected neoplasms (e.g. mesotheliomas and some types of leukemias and lymphomas) could be unterestimated following criticities in the diagnostic process.          

We have included this suggestion and revised it accordingly. Please refer to line 256 – 259.

Lines 242-250: useful to mention other cancers whose classification suffers of a large variability in space and time, such as mesotheliomas, some head and neck cancers, some leukemias and lymphomas; useful to mention that in medium- and low-income countries, the incidence of selected neoplasms (e.g. mesotheliomas and some types of leukemias and lymphomas) could be unterestimated following criticities in the diagnostic process.          

We have incorporated this approach into the analysis commonly observed in epidemiological studies. Please refer to line 256 – 259.

Reviewer 3 Report

Comments and Suggestions for Authors

Congratulations to the authors for providing these notions about the methodological issues need to be considered in studies about the occupation-related cancer incidence. 

All mentioned concerns are of utmost importance in conducting such studies and also upon running a meta-analysis; however, another issue might be considered in interpreting and analyzing studies:

1) Missing data in studies are another aspect of methodological issue that need to be considered in analyzing studies

Comments on the Quality of English Language

The writing quality of manuscript seems fine.

Author Response

Congratulations to the authors for providing these notions about the methodological issues need to be considered in studies about the occupation-related cancer incidence. 

All mentioned concerns are of utmost importance in conducting such studies and also upon running a meta-analysis; however, another issue might be considered in interpreting and analyzing studies:

  • Missing data in studies are another aspect of methodological issue that need to be considered in analyzing studies

We included this factor as a potential explanation for the mixed findings. Please refer to line 53-55

Reviewer 4 Report

Comments and Suggestions for Authors

Thank you very much for submitting your work.

The manuscript is well-written. However, the novelty is very limited in the manuscript. Furthermore, most of the points in the paper, was previously discussed in the available meta-analyses on the topic.

Comments on the Quality of English Language

Quality of the language is acceptable.

Author Response

The manuscript is well-written. However, the novelty is very limited in the manuscript. Furthermore, most of the points in the paper, was previously discussed in the available meta-analyses on the topic.

Thank you